# Determinants of quality antenatal care use in Kenya: Insights from the 2022 Kenya Demographic and Health Survey

**John Baptist Asiimwe**[1]*, **Angella Namulema**[2], **Quraish Sserwanja**[3], **Joseph Kawuki**[4], **Mathius Amperiize**[5], **Earnest Amwiine**[6], **Lilian Nuwabaine**[1]*

1 School of Nursing and Midwifery, Aga Khan University, Kampala, Uganda, 2 Mbarara Regional Referral Hospital, Mbarara, Uganda, 3 Programs Department, Relief International, Khartoum, Sudan, 4 Program in Public Health, Department of Family, Population, & Preventive Medicine, Stony Brook University, Stony Brook, New York, United States of America, 5 Faculty of Medicine, Mbarara University of Science & Technology, Mbarara, Uganda, 6 Infectious Diseases Institute, Kampala, Uganda

* john.asiimwe@aku.edu (JBA); lilliannuwabaine@gmail.com (LN)

**Data Availability Statement:** The Institutional Review Board of the Inner-City Fund (ICF) granted

## Abstract

Provision of quality antenatal care (ANC) is important to reduce maternal and newborn fatalities worldwide. However, the use of quality ANC by women of reproductive age and associated factors remain unclear in many developing countries. Therefore, this study aimed to determine factors associated with receiving quality ANC in Kenya among women of reproductive age. We analyzed secondary data from the 2022 Kenya Demographic Health Survey, which included 11,863 women. Participants were selected using two-stage stratified sampling. Univariate and multivariable logistic regression analyses were used to analyze the data. Of the 11,863 participating women, 61.2% (95% confidence interval (CI): 59.7%–62.6%) received quality ANC. Participants aged 20–34 years had a 1.82 (95%CI: 1.15–2.87) times higher likelihood of receiving quality ANC compared with those aged 15–19 years. Those who had attended four or more ANC visits were 1.42 (95%CI: 1.14–1.79) times more likely to receive quality ANC than those who attended three or fewer visits. Participants with media access were 1.47 (95%CI: 1.06–2.03) times more likely to receive quality ANC than those without media access. Compared with participants in the "poorest" quintile, the likelihood of receiving quality ANC was 1.93 (95%CI: 1.21–3.08) and 1.44 (95% CI: 1.01–2.06) times higher for participants in the "richest" and "richer" quintiles, respectively. Furthermore, compared with participants from the Coastal region, the odds of receiving quality ANC were 0.25 (95%CI: 0.15–0.31) to 0.64 (95%CI: 0.44–0.92) times lower for those from all other Kenyan regions. Participants whose partners made their healthcare decisions were 0.74 (95%CI: 0.58–0.95) times less likely to receive quality ANC than those who made decisions independently. We found that just over 60% of participating mothers had received quality ANC. Factors associated with receiving quality ANC were: age, region, maternal education, healthcare-seeking decision-making, access to media, time to the health facility, ANC visits, and ANC provider type (doctor, nurse/midwife/clinical officer). Maternal health improvement programs should prioritize promoting access to education for girls. Furthermore, interventions should focus on promoting shared decision-making and

ethical approval for the 2022 KDHS. Whereas the Kenya National Bureau of Statistics carried out the study in collaboration with other development partners. Since this study is based on secondary data from the KDHS that is publicly available, no ethical approval was required for its analysis, however, MEASURE DHS provided authorization to use the KDHS datasets (https://www.dhsprogram.com/data/available-datasets.cfm). Both from human participants and from legally appointed representatives of minor participants, written informed consent was acquired.

**Funding:** The authors received no specific funding for this work.

**Competing interests:** The authors have declared that no competing interests exist.

autonomy in healthcare-seeking behaviors among pregnant women and their partners, increasing access to care provided by skilled healthcare workers, and addressing regional disparities in healthcare delivery.

## Introduction

Quality antenatal care (ANC) encompasses assessments and treatments provided by licensed medical practitioners to pregnant women and adolescent girls to maintain the health of the mother and child [1, 2]. Low quality ANC is associated with an increased risk for maternal and newborn mortality and morbidity, including stillbirth, low birth weight, and preterm birth [3]. Quality ANC can prevent maternal mortality by addressing pregnancy-related complications, which emphasizes its significance in improving maternal health outcomes worldwide [4–6]. Sub-Saharan Africa has the highest maternal mortality rates globally. ANC coverage remains low in this region, with only 65% of pregnant women accessing ANC services and a paucity of information about the quality of ANC [3]. In addition, provision of quality ANC is suboptimal in some of these countries (Ethiopia: 31.3%, Nigeria: 45%, and Uganda: 61.4%) [4–6].

Factors influencing women's receipt of quality ANC span socioeconomic, demographic, obstetric, antenatal, and facility-related domains [6–8]. These factors include age, socioeconomic status, education level, partner support, urban residence, media exposure, facility choice, and ANC visit frequency [9–11]. Moreover, early initiation of ANC visits, adherence to the recommended number of visits, and the category of ANC health providers have been linked to higher ANC quality [11, 12]. The maternal mortality rate in Kenya is approximately 342 deaths per 100,000 live births, and the newborn mortality rate is about 19 deaths per 1,000 live births, which indicates there are significant maternal and child health challenges [13]. The Kenyan Ministry of Health has made efforts to improve ANC coverage and quality through various initiatives, including implementation of ANC guidelines, training for healthcare providers, and community sensitization programs [14]. However, the prevalence of quality ANC in Kenya, as defined by adherence to ANC guidelines and the provision of comprehensive care, remains unclear and may vary across facilities and regions [14].

Several previous studies from Kenya examined subnational disparities in ANC use and factors influencing the use of focused ANC services [15, 16]. However, these studies did not provide a comprehensive understanding of how these parameters interacted with and affected quality ANC use at a national level. In addition, the high maternal and newborn mortality rates in Kenya suggest gaps in women's access to quality ANC services during pregnancy. Therefore, we used data from the 2022 Kenya Demographic Health Survey (KDHS) to shed light on the prevalence of quality ANC services and factors that influenced the receipt of these services in Kenya. The results of this study may inform interventions to enhance ANC service delivery and promote positive maternal health outcomes, both in Kenya and sub-Saharan Africa more broadly.

## Methods

### Sampling, data collection, and data source

This study used secondary data from the 2022 KDHS. The KDHS used a two-stage stratified sampling design. The first stage involved selecting 1692 enumeration areas or clusters from a master sample frame of 129,067 clusters based on the 2019 Kenya Population and Housing

Census using equal probability with independent selection [17]. The second stage involved listing houses to generate a sampling frame and choosing 25 households from each cluster. In clusters with fewer than 25 households, all households in that cluster were sampled. The survey was conducted in over 1691 clusters, which gave a nationally representative sample. The training of data collectors and pretesting of study instruments were conducted by the Inner-City Fund (ICF), and data were collected between February and July 2022. Interviews (in Swahili or English) were conducted with all women aged 15–49 years who were regular members of the chosen households or who had spent the night before the survey in that household. Of the 32,156 women who completed the 2022 KDHS, 11,863 women who were either pregnant or had given birth during the previous 5 years were included in the present study. The research team requested and obtained authorization to use the secondary dataset from the MEASURE DHS website (https://www.dhsprogram.com/data/available-datasets.cfm). The dataset contained numerous variables, but the research team selected those that were applicable to this study for inclusion in the analyses.

## Study variables

**Dependent/outcome variable.** Women's receipt of quality ANC was the main outcome variable in this study. ANC quality was a composite variable that was created by combining several binary (yes/no) questions about the provision of certain services during ANC. These services included blood and urine sample collection, blood pressure checks, receiving information about danger signs of pregnancy (e.g., bleeding), fetal heartbeat monitoring, breastfeeding counseling, dietary counseling, and the provision (or purchase) of iron supplements. Participants that had received all eight ANC services were classified as having received quality ANC (yes); if one or more services had been omitted, they were classified as not receiving quality ANC (no) [4, 5].

**Independent variables.** The three categories of covariates considered in this analysis were sociodemographic factors, obstetric and prenatal-related factors, and health facility-related factors, based on a review of the literature and the available KDHS data [4, 5, 17]. The sociodemographic parameters investigated included: education level of the woman and her partner (primary, secondary, or tertiary), the woman's age (15–19, 20–34, or 35–49 years), wealth index (five classes: poorest to richest), place of residence (rural or urban), marital status (single or married), and religion (Christian, Muslim, or others). Region was categorized using Kenya's eight provinces (Nyanza, Western, Eastern, Coast, Northeastern, Central, Rift Valley, and Nairobi). The household size (≤4 or ≥5 people) was used to measure family composition. Two proxy variables were used to assess maternal autonomy: who headed the household (female or male) and who made healthcare-seeking decisions for the participant (partner, self, jointly with another person/partner, or others). We also considered mobile phone ownership (yes or no) and exposure to mass media (i.e., access to newspapers, radio, television, and the Internet; yes or no). Principal component analysis was used to compute the wealth index from data on household asset ownership [17].

Six obstetric and prenatal factors were analyzed: whether the participating woman was currently pregnant, whether she had received information about ANC from a community health worker, parity (≤2, 3–4, ≥5), number of ANC visits (≤3 or ≥4), when she had her first ANC visit (0–3, 4–6, or 7–9 months), and whether the woman had wanted her last pregnancy [17]. We examined five variables that were associated with the location where ANC was received. The place of ANC provision (clinic, faith-based organization, non-governmental organization, private or public health facility), and the person who evaluated the mother during ANC visits (midwife, doctor, clinical officer, nurse, or others). As a proxy measure of access to a health

facility, the number of minutes required to access the health facility for birth ($\leq$30, 31–60, or $\geq$61 minutes) was included in the analysis. An additional proxy measure for participants' familiarity with the healthcare facility was whether or not they had ever taken contraceptives [17].

## Statistical analyses

Data were cleaned and dummy variables were constructed before analysis. For each categorical variable, descriptive statistics (e.g., frequencies) were calculated at the univariate level. We used univariate logistic regression to identify independent variables associated with receiving quality ANC. Simple multivariate logistic regression was then used to identify variables associated with receiving quality ANC while controlling for other variables. All variables with P-values less than 0.05 were included in the multivariate analysis, and 95% confidence intervals (CI) were calculated for all odd ratios. The data were analyzed using the complex samples package in SPSS (V20), which assisted in managing the complicated sample design in the KDHS data. The complex sample package offers reliable parameter estimations as it considers weighting, sample stratification, and clustering throughout the participant sampling process [18]. Furthermore, KDHS sample weights were imposed on all computed frequencies to mitigate the effects of unequal probability sampling in various strata and guarantee the representativeness of the study outcomes [19]. We also evaluated the multi-collinearity of all predictor variables in the model using a variance inflation factor of less than 10 as a cutoff [19]. All predictors fell below this threshold.

## Ethical considerations

The Institutional Review Board of the ICF granted ethical approval for the 2022 KDHS. The Kenya National Bureau of Statistics conducted the survey in collaboration with other development partners. Written informed consent was obtained from all participants or the legally appointed representatives of minor participants. As this study was based on secondary data from the KDHS that are publicly available, no ethical approval was required. However, MEA-SURE DHS provided authorization to use the KDHS datasets (https://www.dhsprogram.com/data/available-datasets.cfm).

## Results

### Participants' demographic characteristics

In total, 11,863 women who were pregnant or had given birth within the 5 years before the survey were included in our analyses (Table 1). The majority identified as being from the Central, Eastern, Rift Valley, Nyanza, and Nairobi provinces (76.7%), were aged 20–34 years (74.4%), and lived in rural areas (61.4%). Most participants were married (80.2%) and most identified as Christian (88.5%). In total, 44.8% had only completed primary (or no) education, 57.3% were employed, and 43.9% were classified in the richer/richest quintiles. Furthermore, most (91%) participants' partners were employed and 55.9% of partners had at least a secondary education. The majority of participants lived in male-headed households (71.5%) and 44.9% made decisions to seek healthcare services jointly with their partner or another person. Most participants lived with their partners (83.2%) and had more than five household members (79%). Many participants were exposed to mass media, which included newspapers (17.2%), television (38.4%), the Internet (47.9%), and radio (74.3%). Furthermore, 81.1% of participants were mobile phone owners. Although most (94%) participants were not currently pregnant,

**Table 1. Participants' demographic characteristics.**

| Variable | n (weighted %) |
|---|---|
| **Age (years)** | |
| 35–49 | 2253 (19.0) |
| 20–34 | 8825 (74.4) |
| 15–19 | 785 (6.6) |
| **Region/province** | |
| Coast | 1107 (9.3) |
| Nairobi | 1371 (11.6) |
| Rift Valley | 3605 (30.4) |
| Northeastern | 406 (3.4) |
| Western | 1253 (10.6) |
| Nyanza | 1406 (11.8) |
| Eastern | 1336 (11.3) |
| Central | 1380 (11.6) |
| **Education** | |
| Tertiary | 2321 (19.6) |
| Secondary | 4231 (35.7) |
| None/primary | 5311 (44.8) |
| **Partner's education** | |
| Tertiary | 2329 (24.5) |
| Secondary | 2984 (31.4) |
| None/primary | 4206 (44.2) |
| **Religion** | |
| Muslim | 1120 (9.7) |
| Christian | 10220 (88.5) |
| Others | 209 (1.8) |
| **Residence** | |
| Rural | 7289 (61.4) |
| Urban | 4574 (38.6) |
| **Wealth index** | |
| Richest | 2695 (22.7) |
| Richer | 2510 (21.2) |
| Middle | 2074 (17.5) |
| Poorer | 2062 (17.4) |
| Poorest | 2523 (21.3) |
| **Marital status** | |
| Married/in a relationship | 9519 (80.2) |
| Unmarried/not in a relationship | 2344 (19.8) |
| **Working status** | |
| Not working | 5063 (42.7) |
| Working | 6791 (57.3) |
| **Partner's working status** | |
| Not working | 858 (9.0) |
| Working | 8632 (91) |
| **Sex of household head** | |
| Female | 3380 (28.5) |
| Male | 8483 (71.5) |
| **Household size** | |

*(Continued)*

**Table 1.** (Continued)

| Variable | n (weighted %) |
| --- | --- |
| ≥5 | 9370 (79.0) |
| ≤4 | 2493 (21.0) |
| **Health seeking decision making** | |
| Joint | 4279 (44.9) |
| Partner | 1590 (16.7) |
| Self | 3618 (38.5) |
| Others | 32 (0.3) |
| **Media access (TV, radio, and newspaper)** | |
| Yes | 1451 (12.2) |
| No | 10412 (87.8) |
| **Mobile phone** | |
| Yes | 9626 (81.1) |
| No | 2237 (18.9) |
| **Internet use** | |
| Yes | 5684 (47.9) |
| No | 6179 (52.1) |
| **Parity** | |
| ≤2 | 6395 (53.9) |
| 3–4 | 3360 (28.3) |
| ≥5 | 2108 (17.8) |
| **Currently pregnant** | |
| Yes | 711 (6.0) |
| No | 11152 (94) |
| **Last pregnancy wanted** | |
| Yes | 10823 (91.2) |
| No | 1041 (8.8) |
| **ANC visits** | |
| ≥4 | 6472 (67.2) |
| ≤3 | 3157 (32.8) |
| **Timing of first ANC visit** | |
| Third trimester | 841 (8.9) |
| Second trimester | 5671 (60.1) |
| First trimester | 2920 (31.0) |
| **Place of ANC** | |
| Public health facility | 7866 (80.5) |
| Private health facility | 1477 (15.1) |
| Faith-based organization | 379 (3.9) |
| Non-governmental organization | 44 (0.5) |
| **ANC provider** | |
| Doctor | 4494 (39.8) |
| Nurse/midwife/clinical officer | 6618 (58.6) |
| Others | 191 (1.6) |
| **Time to health facility (minutes)** | |
| ≥61 | 496 (8.0) |
| 31–60 | 1168 (18.9) |
| ≤30 | 4515 (73.1) |
| **Received ANC service or information from a community health worker** | |

(*Continued*)

**Table 1.** (Continued)

| Variable | n (weighted %) |
|---|---|
| No | 6103 (98.8) |
| Yes | 76 (1.2) |
| **Contraceptive use** | |
| Yes | 6987 (58.9) |
| No | 4876 (41.1) |

Unmarried = single/divorced/widowed/separated.

91.2% had wanted/desired their most recent pregnancy and 53.9% had given birth to two children or had two children currently alive.

The majority of participants (60.1%) had attended their first ANC appointment in the second trimester, and 67.2% had attended at least four ANC appointments overall. ANC visits were most commonly attended at public (83.4%) or private (15.7%) healthcare facilities. Most participants had received ANC from doctors (46.7%) or midwives/nurses/clinical officers (68.7%). Only 1.2% of participants reported learning about ANC from a community health worker and 58.9% had previously used contraceptives. Furthermore, 67.7% had walked to health facilities for ANC, with the most common travel time being 30 minutes (73.1%).

## Quality ANC received

Overall, 61.2% of participants had received quality ANC (**Table 2**). The most commonly received ANC services were blood pressure (98.2%), fetal heartbeat monitoring (97.7%), and blood (97.1%) and urine (95.9%) sample collection. Most (92.2%) participants had also received iron tablets. Sightly fewer participants had received nutritional counseling (84.4%), breastfeeding counseling (82.4%), and counseling about the danger signs of pregnancy (77.0%).

## Factors associated with receiving quality ANC

The factors associated with receiving quality ANC in the univariate and multivariate logistic regression analysis are summarized in **Table 3**. Participants aged 20–34 years had a 1.82 (95% CI: 1.15–2.87) times higher likelihood of receiving quality ANC compared with younger mothers (15–19 years). Participants who had attended four or more ANC visits were 1.42 (95%CI:

**Table 2.** Components of ANC received by participants.

| Variable | weighted % (95%CI) |
|---|---|
| **Overall quality of ANC** | **61.2 (59.7–62.6)** |
| Blood pressure taken | 98.2 (97.9–98.4) |
| Urine samples taken | 95.9 (95.4–96.4) |
| Blood samples taken | 97.1 (96.7–97.5) |
| Fetal heartbeat monitored | 97.7 (97.4–98.1) |
| Nutritional counseling | 84.4 (83.5–85.4) |
| Breastfeeding counseling | 82.4 (81.3–83.4) |
| Advice on danger signs of pregnancy (bleeding) | 77.0 (75.7–78.3) |
| Given/bought iron tablets | 92.2 (91.4–92.8) |

ANC = antenatal care; CI = confidence interval.

**Table 3. Factors associated with receiving quality ANC among women aged 15–49 years.**

| Variable | Quality ANC | | COR (95%CI) | P-value | AOR (95%CI) | P-value |
|---|---|---|---|---|---|---|
| | Yes<br>n (%) | No<br>n (%) | | | | |
| **Age (years)** | | | | <0.001 | | 0.021 |
| 15–19 | 313 (3.3) | 388 (4.0) | 1 | | 1 | |
| 20–34 | 4529 (47.2) | 2650 (27.6) | **2.12 (1.75–2.57)** | | **1.82 (1.15–2.87)** | |
| 35–49 | 1035 (10.8) | 691 (7.2) | 1.86 (1.49–2.33) | | 1.58 (0.94–2.67) | |
| **Residence** | | | | <0.001 | | 0.897 |
| Rural | 3250 (35.4) | 2605 (27.1) | 1 | | 1 | |
| Urban | 2479 (27.5) | 1123 (11.7) | **1.69 (1.47–1.95)** | | 1.017 (0.79–1.31) | |
| **Region/province** | | | | <0.001 | | <0.001 |
| Coast | 602 (6.3) | 251 (2.6) | 1 | | 1 | |
| Northeastern | 81 (0.8) | 204 (2.1) | **0.17 (0.12–.23)** | | **0.25 (0.15–0.31)** | |
| Eastern | 731 (7.6) | 417 (4.3) | **0.73 (0.56–0.95)** | | **0.60 (0.41–0.89)** | |
| Central | 822 (8.6) | 323 (3.4) | **1.06 (0.78–1.44)** | | **0.62 (0.41–0.94)** | |
| Rift Valley | 1587 (16.5) | 1347 (14.0) | **0.49 (0.39–0.61)** | | **0.41 (0.30–0.59)** | |
| Western | 628 (6.5) | 367 (3.8) | **0.71 (0.55–0.93)** | | **0.64 (0.44–0.92)** | |
| Nyanza | 682 (7.1) | 486 (5.1) | **0.58 (0.49–0.75)** | | **0.51 (0.36–0.72)** | |
| Nairobi | 743 (7.7) | 333 (3.5) | **0.93 (0.64–1.36)** | | **0.51 (0.29–0.87)** | |
| **Education** | | | | <0.001 | | 0.098 |
| None/primary | 2293 (23.9) | 1976 (20.6) | 1 | | 1 | |
| Secondary | 2258 (23.5) | 1246 (13.0) | 1.56 (1.37–1.78) | | 1.30 (0.99–1.69) | |
| Tertiary | 1326 (13.8) | 507 (5.3) | **2.25 (1.85–2.75)** | | 1.52 (0.99–2.32) | |
| **Religion** | | | | 0.270 | | |
| Christian | 5117 (54.7) | 3257 (34.8) | 1.31 (0.91–1.89) | | - | |
| Muslim | 480 (5.1) | 336 (3.6) | 1.19 (0.79–1.78) | | - | |
| Others | 88 (0.9) | 73 (0.8) | 1 | | - | |
| **Marital status** | | | | 0.071 | | |
| Married | 4681.912 (48.7) | 2889 (30.1) | 1.14 (0.99–1.31) | | | |
| Unmarried | 1195 (12.4) | 839 (8.7) | 1 | | | |
| **Wealth index** | | | | <0.001 | | 0.001 |
| Poorest | 994 (10.4) | 1071 (11.1) | 1 | | 1 | |
| Poorer | 973 (10.1) | 722 (7.5) | **1.45 (1.24–1.69)** | | 1.19 (0.92–1.53) | |
| Middle | 1043 (10.9) | 662 (6.9) | **1.69 (1.44–1.99)** | | 1.28 (0.97–1.67) | |
| Richer | 1343 (14.0) | 689 (7.2) | **2.09 (1.75–2.52)** | | **1.44 (1.01–2.06)** | |
| Richest | 1523 (15.9) | 584 (6.1) | **2.81 (2.28–3.45)** | | **1.93 (1.21–3.08)** | |
| **Working status** | | | | 0.071 | | |
| Not working | 2463 (25.7) | 1656 (17.3) | 1 | | - | |
| Working | 3411 (35.5) | 2067 (21.5) | 1.11 (0.99–1.24) | | - | |
| **Partner's education** | | | | <0.001 | | 0.635 |
| None/primary | 1875 (24.8) | 1479 (19.5) | 1 | | 1 | |
| Secondary | 1532 (20.2) | 879 (11.6) | **1.38 (1.19–1.58)** | | 0.97 (0.77–1.22) | |
| Tertiary | 1275 (16.8) | 530 (7.0) | **1.89 (1.57–2.29)** | | 0.85 (0.61–0.19) | |
| **Partner's working status** | | | | <0.001 | | 0.153 |
| Not working | 334 (4.4) | 332 (4.4) | 1 | | 1 | |
| Working | 4331 (57.4) | 2549 (33.8) | **1.69 (1.39–2.04)** | | 0.82 (0.62–1.08) | |
| **Sex of household head** | | | | 0.754 | | |
| Male | 4178 (43.5) | 2635 (27.4) | 1 | | - | |

(*Continued*)

**Table 3.** (Continued)

| Variable | Quality ANC | | COR (95%CI) | P-value | AOR (95%CI) | P-value |
|---|---|---|---|---|---|---|
| | Yes n (%) | No n (%) | | | | |
| Female | 1699 (17.7) | 1093 (11.4) | 0.98 (0.87–1.11) | | - | |
| **Health-seeking decision making** | | | | <0.001 | | 0.049 |
| Self | 1797 (23.7) | 1068 (14.1) | 1 | | 1 | |
| Partner | 644 (8.5) | 602 (8.0) | **0.64 (0.54–0.75)** | | **0.74 (0.58–0.95)** | |
| Joint | 2229 (29.4) | 1204 (15.9) | 1.10 (0.96–1.26) | | 1.02 (0.84–1.25) | |
| Others | 12 (0.2) | 16 (0.2) | 0.44 (0.15–1.29) | | 0.49 (0.11–2.22) | |
| **Household size** | | | | <0.001 | | 0.377 |
| ≤4 | 1322 (13.8) | 625 (6.5) | **1** | | 1 | |
| ≥5 | 4554 (47.4) | 3103 (32.3) | **0.69 (0.59–0.81)** | | 0.89 (0.71–1.14) | |
| **Media access (TV, radio, and newspaper)** | | | | <0.001 | | 0.022 |
| Yes | 824 (8.6) | 319 (3.3) | **1.74 (1.39–2.17)** | | **1.47 (1.06–2.03)** | |
| No | 5052 (52.6) | 3409 (35.5) | **1** | | 1 | |
| **Internet use** | | | | <0.001 | | 0.282 |
| No | 2829 (29.5) | 2262 (23.6) | 1 | | 1 | |
| Yes | 3048 (31.7) | 1466 (15.3) | **1.66 (1.48–1.87)** | | 0.87 (0.68–1.12) | |
| **Mobile phone** | | | | <0.001 | | 0.845 |
| No | 993 (10.3) | 872 (9.1) | 1 | | 1 | |
| Yes | 4883 (50.8) | 2856 (29.7) | **1.50 (1.34–1.69)** | | 0.98 (0.79–1.22) | |
| **Parity** | | | | <0.001 | | 0.478 |
| ≤2 | 3274 (34.1) | 1949 (20.3) | 1 | | 1 | |
| 3–4 | 1733 (18.0) | 997 (10.4) | **1.04 (0.92–1.18)** | | 1.06 (0.84–1.34) | |
| ≥5 | 870 (9.1) | 782 (8.1) | **0.66 (0.57–0.78)** | | 1.21 (0.89–1.64) | |
| **Currently pregnant** | | | | 0.121 | | |
| No | 5580 (58.1) | 3509 (36.5) | 1 | | - | |
| Yes | 297 (3.1) | 219 (2.3) | 0.85 (0.69–1.04) | | - | |
| **Last pregnancy wanted** | | | | 0.117 | | |
| No | 502 (5.2) | 364 (3.8) | 1 | | - | |
| Yes | 5375 (56) | 3364 (35) | 1.16 (0.96–1.40) | | - | |
| **Contraceptive use** | | | | <0.001 | | 0.459 |
| No | 2058 (21.4) | 1620 (16.9) | 1 | | 1 | |
| Yes | 3818 (39.8) | 2108 (22.0) | **1.43 (1.28–1.59)** | | 1.08 (0.88–1.32) | |
| **Time to health facility (minutes)** | | | | <0.001 | | 0.068 |
| ≤30 | 2258 (45.5) | 1337 (26.9) | 1 | | 1 | |
| 31–60 | 585 (11.8) | 376 (7.6) | **0.92 (0.77–1.10)** | | 1.24 (1.00–1.52) | |
| ≥61 | 205 (4.1) | 203 (4.1) | **0.60 (0.47–0.77)** | | 0.92 (0.69–1.22) | |
| **ANC visits** | | | | <0.001 | | 0.002 |
| ≤3 | 1572 (16.4) | 1561 (16.3) | 1 | | 1 | |
| ≥4 | 4305 (44.8) | 2167 (22.6) | **1.97 (1.75–2.22)** | | **1.42 (1.14–1.79)** | |
| **Timing of first ANC visit** | | | | <0.001 | | 0.386 |
| First trimester | 2003 (21.2) | 917 (9.7) | 1 | | 1 | |
| Second trimester | 3458 (36.7) | 2213 (23.5) | **0.72 (0.63–0.82)** | | 0.94 (0.76–1.17) | |
| Third trimester | 416 (4.4) | 425 (4.5) | **0.45 (0.37–0.55)** | | 0.77 (0.52–1.12) | |
| **Place of ANC** | | | | | | |
| Public health facility | | | | 0.266 | | |
| No | 1014 (10.7) | 552 (5.8) | 1 | | - | |

(*Continued*)

**Table 3.** (Continued)

| Variable | Quality ANC | | COR (95%CI) | P-value | AOR (95%CI) | P-value |
|---|---|---|---|---|---|---|
| | Yes<br>n (%) | No<br>n (%) | | | | |
| Yes | 4863 (51.6) | 3003 (31.8) | 0.88 (0.71–1.10) | | - | |
| Private health facility | | | | 0.322 | | |
| No | 4920 (52.2) | 3035 (32.2) | 1 | | - | |
| Yes | 956 (10.1) | 520 (5.5) | 1.13 (0.88–1.46) | | - | |
| Faith-based organization | | | | 0.796 | | |
| No | 5637 (59.8) | 3416 (36.2) | 1 | | - | |
| Yes | 240 (2.5) | 140 (1.5) | 0.34 (0.14–0.86) | | - | |
| Non-governmental organization | | | | 0.017 | | 0.386 |
| No | 5861 (62.1) | 3527 (37.4) | 1 | | 1 | |
| Yes | 16 (0.3) | 28 (0.8) | 1.04 (0.77–1.41) | | 1.15 (0.14–0.94) | |
| **ANC provider** | | | | 0.608 | | |
| Skilled | 5756 (59.9) | 3658 (38.1) | 1 | | - | |
| Unskilled | 121 (1.3) | 70 (0.7) | 1.09 (0.78–1.51) | | | |
| **CHW provision of ANC information or service** | | | | 0.611 | | |
| No | 3011 (60.7) | 1895 (38.2) | 1 | | - | |
| Yes | 37 (0.8) | 20 (0.4) | 1.18 (0.63–2.19) | | - | |

**Bold** = significant, * = significant at 0.05, CI = confidence interval, – = not evaluated in that model, Ref. = reference category, COR = crude odd ratio, AOR = adjusted odds ratio, ANC = antenatal care, FBO = faith-based organization, NGO = nongovernmental organization, CHW = community health worker.

1.14–1.79) times more likely to receive quality ANC than those who had attended three or fewer visits. Those with media access were 1.47 (95%CI: 1.06–2.03) times more likely to receive quality ANC than those without media access. Compared with participants in the poorest quintile, the likelihood of receiving quality ANC was 1.93 (95%CI: 1.21–3.08) and 1.44 (95% CI: 1.01–2.06) times higher for those in the richest and richer quintiles, respectively. Compared with the Coastal region, the odds of receiving quality ANC were 0.25 (95%CI: 0.15–0.31) to 0.64 (95%CI: 0.44–0.92) times lower for participants from all other Kenyan regions. Participants whose husbands or partners made their healthcare decisions were 0.74 (95%CI: 0.58–0.95) times less likely to receive quality ANC compared with those who made decisions independently.

## Discussion

This study assessed factors associated with women receiving quality ANC in Kenya using data from the 2022 KDHS. The overall prevalence of receipt of quality ANC among participants was 61.2%. This was higher than the prevalence of quality ANC receipt in Ethiopia (31.38%) [4], Rwanda (13.1%) [19], and Nigeria (45%) [5]. The difference between our study and previous studies may be attributable to differences in health policies and health facility standards, and ANC implementation challenges in various countries [5, 14]. In addition, differential efforts and resources invested in maternal health services, and differences in sociodemographic characteristics across countries may help explain the differences in study findings [1, 4, 20].

Our multivariate analysis showed age was significantly associated with receiving quality ANC. Women aged 20–34 years were more likely to receive quality ANC than women aged 15–19 years. This finding was consistent with studies conducted in Bangladesh and sub-

Saharan Africa, which reported that younger women were less likely to receive quality ANC [2, 8]. Women aged 20–34 years may also be more likely to have better knowledge and understanding of the benefits of quality ANC than younger women because of increased awareness and proactive engagement with the healthcare system, which may increase their demand for and use of such services. We observed significant variations in women receiving quality ANC across different provinces in Kenya. Compared with the Coastal region, women in other regions had lower odds of receiving quality ANC, and those in the North-eastern region had the lowest odds. The Coastal region is near to urban centers and tourist areas of Kenya that have better healthcare facilities [21]. These findings highlighted the presence of regional differences and variations in the provision of ANC, as some regions face challenges such as shortages of skilled healthcare providers, inadequate infrastructure, and limited access to essential resources. Given these regional disparities, it is important that governmental and nongovernmental partners collaborate to address geographical inequities in healthcare delivery.

We found that the wealth index was significantly associated with receiving quality ANC. The richest participants had higher odds of receiving quality ANC compared with the poorest participants. Similar studies in Nigeria also reported that the wealthiest respondents were more likely to adequately use ANC than poorer participants [22, 23]. Rich women may not experience challenges in accessing the money necessary for transport to health facilities and may therefore be able to access more sophisticated healthcare [22]. Empowering Kenyan women by involvement with various development partners through adequate employment/income generating activities should be paramount in policies targeted at optimizing quality ANC use.

Healthcare-seeking decision-making was also associated with quality ANC. Participants whose partners made decisions about their healthcare-seeking were less likely to receive quality ANC compared with those who had joint or independent decision-making. This finding may be partly explained by a qualitative study conducted in Malawi that found most men believed pregnancy and ANC were "women's issues," and therefore perceived decision-making around ANC as low priority [24]. Conversely, women's autonomy in decision-making has a positive effect on their ANC service use [12]. Therefore, interventions by various health and non-health stakeholders that promote shared decision-making and autonomy in healthcare-seeking behaviors among pregnant women and their partners are recommended.

Access to media was significantly associated with quality ANC use in this study. This finding was consistent with studies from Uganda [25] and Bangladesh [2] that established a significant correlation between media exposure and receiving quality ANC. This was because media exposure gave mothers visual and audio access to health-related information, which eventually improved access to healthcare and demand for ANC services [26]. We recommend that more reviews by researchers are conducted to establish the most effective media type to promote uptake of quality ANC.

In this study, the number of ANC visits attended was significantly associated with receiving quality ANC. Women who had attended four or more ANC visits were more likely to receive quality ANC compared with those who had attended fewer visits. This finding was consistent with an Ethiopian study [1] that found pregnant women who had visited hospitals for ANC four or more times had high odds of receiving quality ANC. Attending ANC more than four times increases the chances of obtaining multiple ANC services, including identifying complications and risky behaviors during pregnancy, and is an important indicator of the quality ANC received [3].

### Strengths and limitations

We used the most recent available data from the 2022 KDHS. The large sample size and rigorous KDHS data collection protocols mean the results of our study are generalizable to Kenya and more broadly to sub-Saharan Africa. However, this study was based on secondary data provided by KDHS respondents, which may be subject to recall bias. In addition, the cross-sectional design of this study allowed inferences regarding association but not causality.

### Conclusion

This study revealed that 61.2% of participating women had received quality ANC. We identified several factors associated with receiving quality ANC, including age, region, wealth index, health-seeking decision making, access to media, and ANC visits. Therefore, interventions by various health and non-health stakeholders that promote shared decision-making and autonomy in healthcare-seeking behaviors among pregnant women and their partners are recommended. The regional disparities observed in this study highlight the importance of addressing geographical inequities in healthcare delivery by various governmental and nongovernmental partners. We recommend that more reviews by researchers are conducted to establish the most effective media for quality ANC use.

### Acknowledgments

We are grateful that the data used in this investigation were made available by the Demographic Health Survey program. The authors express sincere gratitude to Ms. Audrey Holmes who copyedited a draft of this manuscript.

### Author Contributions

**Conceptualization:** John Baptist Asiimwe, Lilian Nuwabaine.

**Formal analysis:** John Baptist Asiimwe, Angella Namulema, Quraish Sserwanja, Joseph Kawuki, Mathius Amperiize, Earnest Amwiine, Lilian Nuwabaine.

**Investigation:** Lilian Nuwabaine.

**Methodology:** John Baptist Asiimwe, Angella Namulema, Quraish Sserwanja, Joseph Kawuki, Mathius Amperiize, Earnest Amwiine, Lilian Nuwabaine.

**Project administration:** John Baptist Asiimwe.

**Validation:** Quraish Sserwanja, Joseph Kawuki.

**Writing – original draft:** John Baptist Asiimwe, Angella Namulema, Quraish Sserwanja, Joseph Kawuki, Mathius Amperiize, Earnest Amwiine, Lilian Nuwabaine.

**Writing – review & editing:** John Baptist Asiimwe, Angella Namulema, Quraish Sserwanja, Joseph Kawuki, Mathius Amperiize, Earnest Amwiine, Lilian Nuwabaine.

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
