## [Decision Letter · Decision Letter 0]

10 Jul 2024

PGPH-D-24-01409

Determinants of quality Antenatal Care utilization in Kenya: insights from the 2022 Kenya Demographic and Health Survey

Dear Dr. Asiimwe,

Thank you for submitting your manuscript to PLOS Global Public Health. After careful consideration, we feel that it has merit but does not fully meet PLOS Global Public Health’s publication criteria as it currently stands. Therefore, we invite you to submit a revised version of the manuscript that addresses the points raised during the review process.

We look forward to receiving your revised manuscript.

Kind regards,

Dvora Joseph Davey

Academic Editor

Journal Requirements:

Additional Editor Comments (if provided):

Reviewers' comments:

Reviewer's Responses to Questions

**Comments to the Author**

1. Does this manuscript meet PLOS Global Public Health’s publication criteria? Is the manuscript technically sound, and do the data support the conclusions? The manuscript must describe methodologically and ethically rigorous research with conclusions that are appropriately drawn based on the data presented.

Reviewer #1: Yes

Reviewer #2: Yes

2. Has the statistical analysis been performed appropriately and rigorously?

Reviewer #1: Yes

Reviewer #2: Yes

3. Have the authors made all data underlying the findings in their manuscript fully available (please refer to the Data Availability Statement at the start of the manuscript PDF file)?

Reviewer #1: Yes

Reviewer #2: Yes

4. Is the manuscript presented in an intelligible fashion and written in standard English?

Reviewer #1: Yes

Reviewer #2: Yes

5. Review Comments to the Author

Reviewer #1: Overall, this paper was well written paper and the authors have demonstrated good knowledge of the subject matter. The authors should do well to fix the issues raised to help improve the overall quality of the paper.

Reviewer #2: The findings highlight persistent gaps in ANC quality despite efforts by the Ministry of Health, underscoring the need for targeted interventions to enhance healthcare provider training, community awareness, and adherence to ANC guidelines to improve maternal and newborn health outcomes.

1. the authors need to edit the grammar

2. The authors need to revise the keywords and make them meaningful

3."Ethiopia having a prevalence of 31.3%, Uganda at 61.4%, and Nigeria at 45%" please put this in order of percentages

4.line 108, explain what ICF is

5. explain about the study setting

6, The authors need to make their discussion more stronger, it doesnt fully discuss the results

6. PLOS authors have the option to publish the peer review history of their article (what does this mean?). If published, this will include your full peer review and any attached files.

**Do you want your identity to be public for this peer review?** For information about this choice, including consent withdrawal, please see our Privacy Policy.

Reviewer #1: **Yes: **BOTHA, Nkosi Nkosi

Reviewer #2: No

---

## [Editor Report · Decision Letter 1]

12 Aug 2024

Determinants of quality antenatal care use in Kenya: insights from the 2022 Kenya Demographic and Health Survey

PGPH-D-24-01409R1

Dear Mr., Asiimwe,

We are pleased to inform you that your manuscript 'Determinants of quality antenatal care use in Kenya: insights from the 2022 Kenya Demographic and Health Survey' has been provisionally accepted for publication in PLOS Global Public Health.

Best regards,

Dvora Joseph Davey

Academic Editor